# A Novel Competitive ELISA for Specifically Measuring and Differentiating Immune Responses to Classical Swine Fever C-Strain Vaccine in Pigs

**DOI:** 10.3390/v14071544

**Published:** 2022-07-15

**Authors:** Lihua Wang, Shijiang Mi, Rachel Madera, Yuzhen Li, Wenjie Gong, Changchun Tu, Jishu Shi

**Affiliations:** 1Department of Anatomy and Physiology, College of Veterinary Medicine, Kansas State University, Manhattan, KS 66506, USA; rachelmadera@vet.k-state.edu (R.M.); yuzhen@vet.k-state.edu (Y.L.); 2Key Laboratory of Zoonoses Research, Ministry of Education, College of Veterinary Medicine, Jilin University, Changchun 130012, China; xtyzyb@126.com (S.M.); gwj020406@163.com (W.G.); changchun_tu@hotmail.com (C.T.); 3Changchun Veterinary Research Institute, Chinese Academy of Agricultural Sciences, Changchun 130012, China

**Keywords:** classical swine fever (CSF), C-strain, E^rns^, competitive ELISA (cELISA), monoclonal antibody, DIVA

## Abstract

Classical swine fever can be controlled effectively by vaccination with C-strain vaccine. In this study, we developed a novel competitive enzyme-linked immunosorbent assay (cELISA) based on a C-strain E^rns^ specific monoclonal antibody (mAb 1504), aiming to serologically measure immune responses to C-strain vaccine in pigs, and finally to make the C-strain become a DIVA-compatible vaccine. The cELISA system was established based on the strategy that mAb 1504 will compete with the C-strain induced antibodies in the pig serum to bind the C-strain E^rns^ protein. The cELISA was optimized and was further evaluated by testing different categories of pig sera. It can efficiently differentiate C-strain immunized from wild-type CSFV-infected pigs and lacks cross-reaction with other common swine viruses and viruses in genus *Pestivirus* such as Bovine viral diarrhea virus (BVDV). The C-strain antibody can be tested in pigs 7–14 days post vaccination with this cELISA. The sensitivity and specificity of the established cELISA were 100% (95% confidence interval: 95.60 to 100%) and 100% (95% confidence interval: 98.30 to 100%), respectively. This novel cELISA is a reliable tool for specifically measuring and differentiating immune responses to C-strain vaccine in pigs. By combining with the wild-type CSFV-specific infection tests, it can make the C-strain have DIVA capability.

## 1. Introduction

Classical swine fever (CSF) is a highly contagious viral disease of domestic and wild pigs [1]. Despite the enormous control efforts, it continues to cause significant economic losses to the swine industry worldwide and represents a high-consequence threat to agriculture security and trade for CSF-free countries such as the United States [2,3,4]. The conventional Chinese vaccine (C-strain) is the most frequently used live attenuated vaccine for CSF control and prevention [5]. It was attenuated from a virulent strain over at least 480 passages in rabbits [6]. The immunity has been proven to persist for at least six to eleven months, probably even lifelong. It can cover all different genotypes, does not require adjuvants, and is suitable for the oral vaccination of wild boar populations [7,8,9]. The only drawback of the C-strain vaccine is the lack of a reliable accompanying diagnostic assay that allows the differentiation of infected from vaccinated animals (DIVA), which has hindered its application in the control and elimination of CSF outbreaks, especially in CSF-free countries. 

CSF is caused by the classical swine fever virus (CSFV), a small enveloped single stranded positive-sense RNA virus that belongs to the genus *Pestivirus* in the family *Flaviviridae* [1]. CSFV exists as a single serotype and has evolved into three distinct genotypes and eleven sub-genotypes based on phylogenetic analysis with E2, 5′ UTR, or NS5B gene sequences [10,11,12]. The viral envelope glycoprotein E^rns^ is one of the major targets for eliciting antibodies against CSFV in infected animals [13,14]. It has been shown that antibodies to E^rns^ can be used as an indicator of CSFV infection in pigs. The E^rns^-based enzyme-linked immunosorbent assay (ELISA) can be used as a companion diagnostic test to identify CSFV-infected pigs vaccinated with the E2-based subunit or marker vaccines [15,16,17,18,19]. Here, we describe a competitive ELISA (cELISA) developed with a C-strain E^rns^ specific monoclonal antibody (mAb), which can specifically measure and differentiate immune responses to C-strain vaccine in pigs. 

## 2. Materials and Methods

### 2.1. Animals

Five female Balb/c mice (six weeks old) were purchased from Charles River Laboratories, Inc. Wilmington, MA, USA. The mice were fed with a standard commercial diet and housed in a clean facility at Kansas State University. Animal care and protocols were approved by Institutional Animal Care and Use Committee (IACUC#4490) at Kansas State University. All animal experiments were performed under strict adherence to the IACUC protocols. 

### 2.2. Cell Lines and Media

*Spodoptera frugiperda* (Sf9; ATCC, Manassas, VA, USA) and High Five (ATCC, Manassas, VA, USA) insect cells were grown at 27 °C under an air atmosphere in Grace’s insect medium (Gibco, New York, NY, USA) supplemented with 10% fetal bovine serum (FBS; Atlanta Biologicals, Flowery Branch, GA, USA) and 1% antibiotic-antimycotic solution (Gibco, New York, NY, USA), and Express Five SFM medium (Gibco, New York, NY, USA), respectively. Murine myeloma cell line Sp2/0Ag14 was purchased from the American Type Culture Collection (ATCC-CRL-1581, Rockville, MD, USA) and was maintained in RPMI-1640 (Gibco, New York, NY, USA) supplemented with 10% FBS at 37 °C with 5% CO_2_.

### 2.3. Serum Samples

The negative control sera from phosphate-buffered saline (PBS) inoculated pigs (*n* = 159), C-strain/ C-strain E^rns^ immunized pig sera (*n* = 45), C-strain E2 immunized pig sera (*n* = 151), CSFV Alford (Genotype 1.1)-infected pig sera (*n* = 223), CSFV Honduras/1997 (Genotype 1.3)-infected pig sera (*n* = 59), porcine reproductive and respiratory syndrome virus (PRRSV)-infected pig sera (*n* = 107) are kept in our laboratory at Kansas State University. Pseudorabies virus (PRV)-infected pig serum sample (*n* = 1) and BVDV-infected pig serum samples (*n* = 2) were purchased from National Veterinary Services Laboratories (NVSL, Ames, IA, USA). The existence of CSFV E^rns^ antibodies in these serum samples were tested by ELISA as we described previously [20]. Briefly, 62.5 ng/mL of purified Erns was used as coating antigen on 96-well flat-bottomed microtiter plates (Corning^®^, ARI, Corning, NY, USA). Diluted sera (1:2000) were added to plates and incubated for 1 h (hr) at room temperature. Then, horseradish peroxidase (HRP)-conjugated goat anti-porcine IgG was used as secondary antibody (Southern Biotech, Birmingham, AL, USA). The ELISA plates were developed using 3,3,5,5 tetramethylbenzidine (TMB) stabilized chromogen (Thermo Scientific, Rockford, IL, USA), and the reactions were stopped with 2 N sulfuric acid. Relative antibody concentrations were determined using optical spectrophotometer readings at 450 nm using a SpectraMAX microplate reader. 

### 2.4. Expression of C-Strain E^rns^ Protein and Generation of Monoclonal Antibodies

Expression and purification of C-strain E^rns^ protein using a baculovirus expression system were performed as we previously described [20]. Briefly, PCR amplified E^rns^ gene from hog cholera lapinized virus C-strain (HCLV, Genotype 1.1) was cloned into pFastBac^TM^ 1 plasmid and transformed into DH10Bac^TM^
*E. coli* host strain (Thermo Scientific, IL, USA). To generate recombinant baculovirus stock for E^rns^ expression, Sf9 insect cells were transfected with the bacmid DNA extracted from the DH10Bac^TM^
*E. coli* and passaged three times to amplify the E^rns^ bearing recombinant baculovirus. At passage 3, Sf9 cell culture supernatant was collected and clarified by centrifugation at 500× *g* for 5 min to obtain the baculovirus stock that was used to infect High Five™ Cells for E^rns^ expression. E^rns^ protein was purified using Ni-NTA Agarose (Thermo Scientific, IL, USA) as described by the manufacturer. The purified C-strain E^rns^ protein was concentrated using Amicon Ultra Centrifugal Filters 30,000 NMWL (Millipore, Billerica, MA, USA) and measured using BCA assay kit (Thermo Scientific, IL, USA) according to the manufacture’s recommendations. 

Generation of monoclonal antibodies against C-strain E^rns^ were performed as we previously described [21]. Briefly, 50 µL (1 µg/µL) purified E^rns^ protein plus equal volume of Alhydrogel 2% (InvivoGen, San Diego, CA, USA) was used as an immunogen to inject each of five female Balb/c mice (purchased from Charles River Laboratories, Inc. Wilmington, MA, USA) via intraperitoneal injection for mAb production. Three booster immunizations with same dose were conducted at two-week intervals. Three days after the final booster injection, the mice were euthanized and spleen cells were fused with the mouse myeloma partner SP2/0-Ag14 (ATCC, Gaithersburg, MD, USA) by using polyethylene glycol 1500 (Roche Diagnostics, Indianapolis, IN, USA) at a ratio of 10:1. The hybridoma cells were maintained in RPMI1640 medium (Gibco, New York, NY, USA) with 20% FBS. Supernatants from growing hybridomas were screened by an ELISA for reactivity to E^rns^ protein. The positive hybridoma clones were subcloned three times by limiting dilution until monoclonals were obtained. Isotypes of the generated mAbs were determined with an antibody-isotyping kit (Roche Diagnostics, Indianapolis, IN, USA). 

### 2.5. SDS-PAGE and Western Blotting

For analyzing the insect cell expressed C-strain E^rns^ protein, the purified protein was treated without (Native) or with β-mercaptoethanol (Reduced), separated by SDS-PAGE in a Mini-Protean TGX Gel (Bio-Rad, Hercules, CA, USA) and stained with PageBlue Protein Staining Solution (Thermo Scientific, IL, USA) according to the manufacture’s recommendations. 

For analyzing if the monoclonal antibody recognizes linear or conformational epitope, E^rns^ proteins (native or reduced) were separated by electrophoresis in 10% polyacrylamide gels, and transferred to PVDF membranes (Millipore, Burlington, MA, USA). Membranes were blocked with 5% milk and then incubated with monoclonal antibodies. Incubation with horseradish peroxidase (HRP) conjugated secondary antibody, and detection and imaging were performed as we described previously [20]. 

### 2.6. Competitive Enzyme-Linked Immunosorbent Assay (cELISA)

The mAb1504 was purified by HiTrap™ Protein G column (GE Healthcare Life Sciences, Pittsburgh, PA, USA) followed by conjugating with horseradish peroxidase (HRP) using EZ-Link™ Plus Activated Peroxidase (Thermo Scientific, Bridgewater, NJ, USA) according to the manufacturer’s instruction. The HRP-1504 was dialyzed with Slide-A-Lyzer Dialysis Cassettes (Thermo Scientific, Bridgewater, NJ, USA) against PBS and stored in Pierce™ Peroxidase Conjugate Stabilizer (Thermo Scientific, Bridgewater, NJ, USA). 

Optimization of the concentration of capture antigen and HRP-1504, the dilution of serum, and the blocking solution was performed as we described previously [21]. Briefly, Corning^®^ 96 Well Clear Flat Bottom Polystyrene High Bind Microplates (Corning, NY, USA) were coated overnight with C-strain E^rns^ (0.31 µg/mL, 100 µL/well) in PBS (without calcium and magnesium, pH7.4, Thermo Scientific, Bridgewater, NJ, USA) at 4 °C. After blocking, 50 µL of diluted serum samples and 50 µL of diluted HRP-1504 (0.625 µg/mL) were added to each well and mixed well by pipetting. After adding the TMB Stabilized Chromogen (Invitrogen, Carlsbad, CA, USA) and 2N Sulfuric Acid (Ricca Chemical Company, Arlington, TX, USA), the optical density at 450 nm (OD_450_) were obtained using SpectraMAX microplate reader (Molecular Devices, San Jose, CA, USA). The OD_450_ of the samples were converted to a percent inhibition (PI) value using the following formulation: PI (%) = (OD_450_ value of negative controls − OD_450_ value of sample)/OD_450_ value of negative controls × 100%. 

### 2.7. Reproducibility and Statistical Analysis 

Inter-assay and intra-assay reproducibility for the established cELISA was evaluated by testing C-strain E^rns^ antibody negative (*n* = 20) and C-strain E^rns^ positive (*n* = 20) pig serum samples. For the intra-assay reproducibility, each serum sample (in duplicate) was detected by the same batch of pre-coated ELISA plates. For the inter-assay reproducibility, each serum sample was detected by three batches of pre-coated ELISA plates. Sensitivity and specificity analysis were carried out by the web-based MedCalc statistical software (https://www.medcalc.org/calc/diagnostic_test.php, accessed on 30 May 2022). Statistical analysis of reproducibility was carried out by calculating the mean PI value and coefficient of variation (CV) of replications of each test.

## 3. Results

### 3.1. Generation of Capture Antigen and Suitable Competitive Monoclonal Antibody

The envelope glycoprotein E^rns^ of C-strain was successfully expressed in insect cells by using Bac-to-Bac^®^ Baculovirus Expression System. The purified C-strain E^rns^ protein exists as homodimer and monomer under non-reducing condition with a molecular weight of ~74 kDa and ~37 kDa, respectively (Figure 1A).

To generate suitable competitive mAb, the purified C-strain E^rns^ protein was used as an immunogen in Balb/c mice. After three injections and the fusion of spleen cells with myeloma cells, we generated one panel of more than ten hybridomas which secreting mAbs against C-strain E^rns^ protein. After further evaluation by Western blot, mAb 1504 (IgG1 and kappa chain) which recognizes the conformational epitope of C-strain E^rns^ (Figure 1B) showed the best of capability to differentiate C-strain from wild-type CSFVs. It can only recognize insect cell expressed C-strain E^rns^, but not react with insect cell expressed E^rns^ of wild-type CSFVs which are prevalent in Europe (sub-genotypes 2.1 and 2.3) [22], Asia (sub-genotypes 1.1 and 2.1–2.3) [23], and other separated geographic regions (sub-genotypes 3.1, 3.2 and 3.4) [22,23,24]. Indirect fluorescent antibody assay (IFA) testing further confirmed that mAb 1504 lacks reaction to CSFV field isolates (genotypes 1.1, 2.1a, 2.1b, 2.1c, 2.1g, 2.1h, 2.1i, 2.1j, 2.2 and 2.3). Detailed data are described in our article for characterization of mAbs that specifically differentiate field isolates from vaccine strains of CSFV [25].

### 3.2. Establishment of Competitive ELISA System with C-Strain E^rns^ Protein and mAb 1504

The development and optimization of the cELISA system were performed as described in our previous publication [21]. A concentration of 0.31 µg/mL of C-strain E^rns^ protein and a concentration of 0.625 µg/mL of HRP-1504 were chosen for optimal concentrations of capture protein and competitive mAb, because they consistently produced an OD_450_ value around 2.2 and at a point in the linear range in the standard curve (Figure 2A); 2% FBS in PBST was chosen as the optimal blocking buffer. The preferred dilution of serum samples is 1:5 as it is the lowest dilution with the highest differential ability between negative pig sera and C-strain-vaccinated pig sera (Figure 2B). These conditions were used in all subsequent cELISA experiments. 

### 3.3. Standardization of the Cut-Off Value of the Established cELISA

A total of 549 pig serum samples were used to standardize the cut-off value of the established cELISA. Among them, 504 samples were from unvaccinated negative control pigs and 45 samples were from C-strain or C-strain E^rns^ protein immunized pigs (14–56 days post vaccination, DPV). Distributions of the cELISA PI values showing the frequency of positive and negative samples are calculated and shown in Figure 3. The mean PI value (x-axis) of the negative sera detected by cELISA is −0.12%. When the mean PI value of negative sera plus two standard deviation (SD) (i.e., 10.63%) was used as threshold, the sensitivity and specificity of the cELISA were 100% (95% confidence interval: 95.60 to 100%) and 97.90% (95% confidence interval: 95.70 to 99.10%), respectively. When mean PI value of negative sera plus three SD (i.e., 15.99%) was used as threshold, the sensitivity and specificity of the cELISA were 100% (95% confidence interval: 95.60 to 100%) and 100% (95% confidence interval: 98.30 to 100%), respectively. 

### 3.4. The Established cELISA Can Efficiently Differentiate C-Strain/C-Strain E^rns^ Immunized Pigs from Wild-Type CSFV and Other Common Swine Virus-Infected Pigs

For validating the specificity of the established cELISA, we tested eight categories of serum samples including negative control sera from PBS inoculated pigs (*n* = 159), C-strain/C-strain E^rns^ immunized pig sera (*n* = 37, 14–56 DPV), C-strain E2 immunized pig sera (*n* = 151, 0–52 DPV), CSFV Alford (Genotype 1.1)-infected pig sera (*n* = 223, 0–15 days post infection, DPI), CSFV Honduras/1997 (Genotype 1.3)-infected pig sera (*n* = 59, 0–15 DPI), PRRSV-infected pig sera (*n* = 107, 0–38 DPI), PRV-infected pig sera (*n* = 1) and BVDV-infected pig sera (*n* = 2). The negative control and CSFV E2 immunized pig sera showed PI value from −19.59 % to 8.26%. The wild-type CSFVs Alford and Honduras/1997-infected pig sera showed PI value from −19.12% to 8.10%. The other common swine viruses-infected pig sera showed PI value from −14.93% to 3.69%. However, all C-strain/C-strain E^rns^ immunized pig sera showed PI value from 23.82% to 92.03% (Figure 4). By using mean PI value (−0.32%) of negative sera plus three SD (i.e., 16.39%) as a threshold, this cELISA can efficiently differentiate C-strain/C-strain E^rns^ immunized pig sera from the other seven pig sera categories. 

### 3.5. Reproducibility of the cELISA

The reproducibility of the cELISA was determined by calculating the coefficient of variation (CV) of PI values by testing 20 C-strain E^rns^ negative pig serum samples and 20 C-strain E^rns^ positive pig serum samples. The intra-assay CVs of the C-strain E^rns^ positive samples ranged from 0.05 to 1.37%. The inter-assay CVs of those same samples ranged from 0.86 to 8.50%. The intra-assay and inter-assay of the C-strain E^rns^ negative samples also exhibited excellent repeatability, showing 0.10–0.45% and 0.56–5.45%, respectively (Table 1).

### 3.6. Kinetics of E^rns^ Antibody Response of C-Strain-Vaccinated Pigs at Different Time Intervals 

Serial serum samples from 0 to 56 DPV at every 7 days were derived from three pigs vaccinated with C-strain and were tested by the established cELISA. The C-strain E^rns^ antibody could be detected in all three pigs between 7 to 14 DPV. Significant increase of inhibition was observed between 7 DPV and 21 DPV. The levels of antibody titer kept relatively stable from 28 DPV to 56 DPV with inhibition values ranging from 62.81% to 84.14% (Figure 5).

## 4. Discussion

Vaccines with DIVA capability with their accompanying diagnostic tests are essential to control and to enabling a country’s rapid return to free status following a CSF outbreak. Up to now, only one CSF DIVA vaccine was licensed by the European Medicines Agency: “CP7_E2alf” (Alfort/187 E2-based “CP7_E2alf” chimeric pestivirus, Suvaxyn CSF Marker, Zoetis) [26]. Its DIVA concept is based on the fact that field-virus-infected animals react positive in an E^rns^-specific antibody ELISA, while vaccinated animals only develop a CSFV E2-specific antibody response. Two ELISAs are commercially available that could work with CP7_E2alf DIVA vaccine. One is the PrioCHECK CSFV E^rns^ (Thermofisher, former Prionics), and the other is the pigtype CSF Marker (Qiagen, Germantown, MD, USA). However, both of them showed limited specificity and sensitivity [18,19,27]. Therefore, novel reliable DIVA diagnostic assays, which can accompany current and/or future CSF vaccines, are critically needed. 

C-strain is the most frequently used vaccine all over the world and showed outstanding efficacy and safety from many years of use [2,3,4,5,6,7,8,9]. A single vaccination with a C-strain-based vaccine can provide lifelong immunity with an onset as early as 2–3 days after vaccination [5]. The only disadvantage of the C-strain vaccine is that it does not allow for DIVA, which can lead to trade restrictions with significant economic consequences. As alternatives, virus replicon particles (VRP) were developed with deletions of E^rns^, E2, or partial E2 sequences of C-strain to make it compatible with a DIVA approach to serology [28,29]. However, it has been reported that the use of this type of vaccine is limited depending on the replicon type and administration route. This caused differences in the level of protection of pigs against the challenge of virulent CSFVs [9,28,29]. 

In this study, we successfully generated a C-strain-specific mAb 1504, which can only recognize E^rns^ protein of C-strain, but not react with all prevalent CSFV strains in different genotypes. This unique character combined with lacking cross-reaction with other antigenically closely related pestiviruses make mAb 1504 an ideal competitive antibody to develop a C-strain-specific cELISA. The hypothesis of the C-strain-specific cELISA is that mAb 1504 can compete with the C-strain induced antibodies in the pig serum to bind C-strain E^rns^ protein. cELISA assays have been shown to increase specificity and reduce the cross-reactivity [30,31]. Indeed, after optimizing, our established cELISA can efficiently differentiate C-strain immunized from wild-type CSFVs or other common swine virus-infected pigs and can test the C-strain antibody during 7–14 days post vaccination. Both sensitivity and specificity can reach 100% when using mean PI value of negative sera plus three SD as threshold. By combining this C-strain-specific cELISA with the wild-type CSFV specific infection tests, such as the genetic detection systems developed to specifically detect wild-type CSFVs [32,33,34,35,36,37], it can differentiate the infected (C-strain-specific C-ELISA “-”, wild-type CSFV specific infection tests “+”) from C-strain-vaccinated pigs (C-strain-specific C-ELISA “+”, wild-type CSFV specific infection tests “-”), and it can also be used to detect the infected pigs that are incompletely protected by C-strain vaccination (C-strain-specific C-ELISA “+”, wild-type CSFV specific infection tests “+”) as well. This will play a critical role in making decisions for a control strategy that employs vaccination with C-strain.

## 5. Conclusions

The presented mAb 1504-based cELISA showed excellent specificity and sensitivity. This cELISA is a reliable and cost-effective tool for specifically measuring and differentiating immune responses to the C-strain vaccine in pigs. By combining with the wild-type CSFV-specific infection tests, it renders the C-strain vaccine DIVA compatible. 

## Figures and Tables

**Figure 1 viruses-14-01544-f001:**
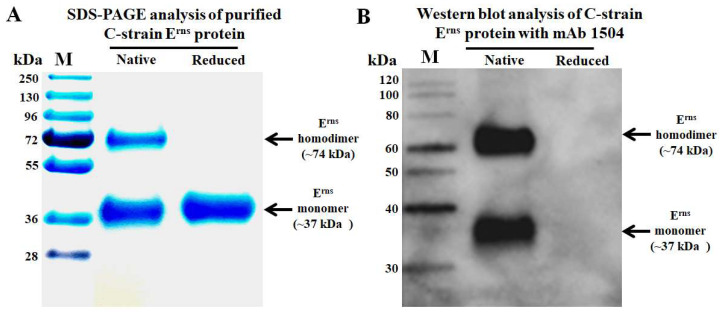
Analysis of purified C-strain E^rns^ protein and monoclonal antibody 1504. (**A**) Purified C-strain E^rns^ protein exists as homodimer and monomer under native condition. (**B**) mAb 1504 recognizes the conformational epitope as it only reacts with the native C-strain E^rns^ protein. The left lines are protein molecular weight size marker.

**Figure 2 viruses-14-01544-f002:**
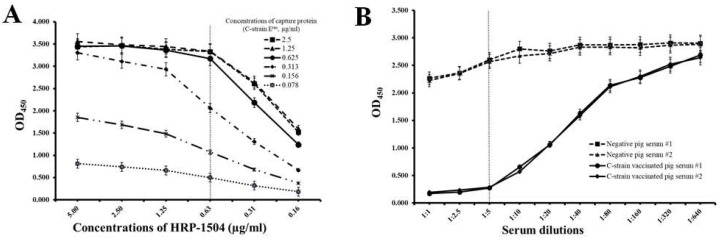
Determination of concentrations of capture antigen and competitive antibody, and dilution of serum. (**A**) Determination of optimal concentrations of capture protein (C-strain E^rns^) and competitive antibody (HRP-1504). (**B**) Determination of an optimal dilution of serum. Data are expressed as the mean ± standard deviation from independently repeated experiments.

**Figure 3 viruses-14-01544-f003:**
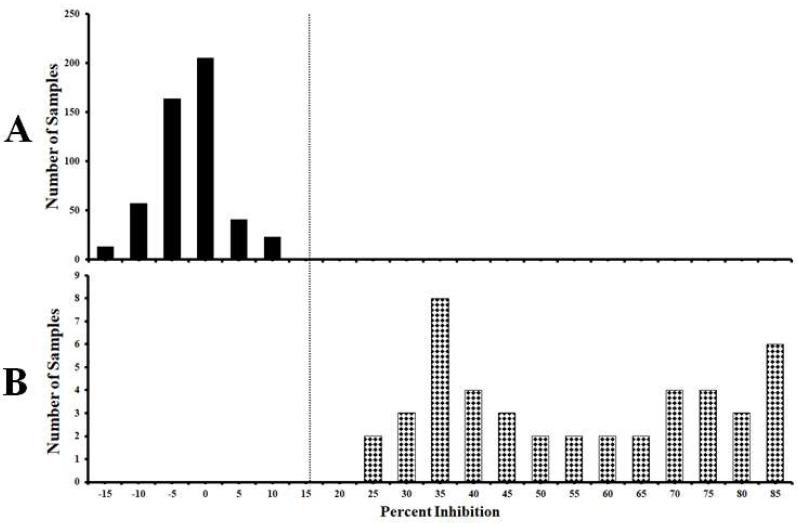
Standardization of the cut-off value of mAb1504-based cELISA. (**A**) Negative serum samples from unvaccinated pigs (*n* = 504). (**B**) C-strain antibody positive serum samples from C-strain or C-strain E^rns^ protein immunized pigs (14–56 DPV) (*n* = 45). The dotted line represents cut-off value of 15.99% inhibition when using the mean PI of negative sera plus three SD as the threshold.

**Figure 4 viruses-14-01544-f004:**
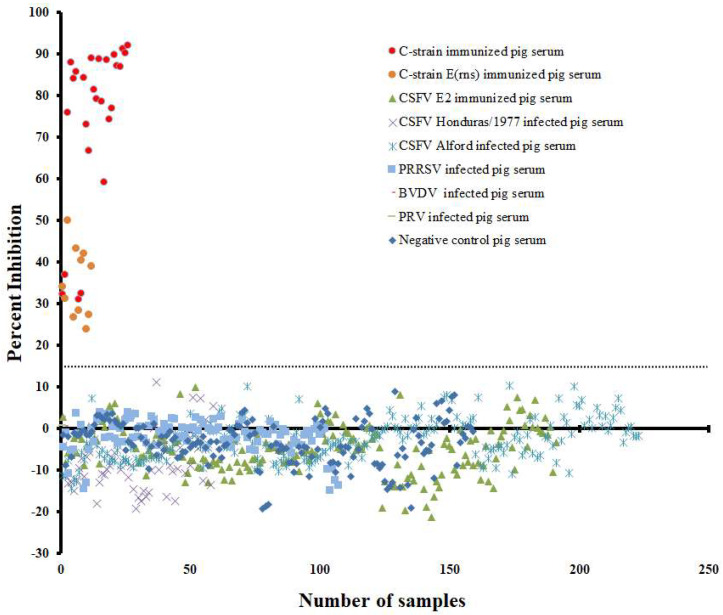
Validating the diagnostic specificity of the mAb1504-based cELISA. The dotted line represents cut-off value of 16.39% inhibition when using the mean PI of negative sera plus three SD as the threshold.

**Figure 5 viruses-14-01544-f005:**
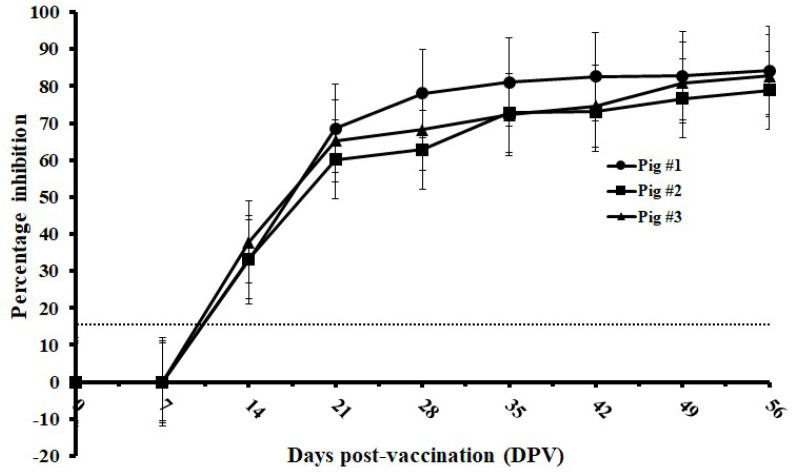
Kinetics of E^rns^ antibody response of C-strain-vaccinated pigs tested by mAb1504-based cELISA. Serum samples were derived from C-strain-vaccinated pigs (*n* = 3) at every 7 days. The dotted line represents cut-off value: 16%.

**Table 1 viruses-14-01544-t001:** Coefficient values of the samples tested by mAb1504-based cELSIA.

Inhibition Range (%)	No. of Serum Tested	CV Range (%)	
		Intra-Assay	Inter-Assay
65–80	20	0.05–1.37	0.86–8.50
<8	20	0.10–0.45	0.56–5.45

## Data Availability

The datasets for the current study are available from the corresponding author on reasonable request.

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
