# Peer review of "A Novel Competitive ELISA for Specifically Measuring and Differentiating Immune Responses to Classical Swine Fever C-Strain Vaccine in Pigs"

_viruses, 2022, doi:10.3390/v14071544_

Round 1

Reviewer 1 Report

In this manuscript, Jishu et al. described the a competitive ELISA (cELISA) developed with a C-strain Erns specific monoclonal antibody (mAb), which can specifically measure and differentiate immune

responses to C-strain vaccine in pigs. Firstly, they generated the antigen and obtained the effective competitive monoclonal antibody, then showed the cross-reactivity of mAb with CSFV field isolates and other viruses in genus Pestivirus. Finally, they used the optimized competitive ELISA to evaluated vaccinated pigs. Overall, it is an exciting and complete story that highlights the elaborate arms race serologically measuring immune responses to C-strain vaccine in pigs, and making the C-strain become a DIVA compatible vaccine. I think it will be exciting for the broad readership of this journal due to the application of the mAb in research of CSFV.

Below is a list of suggestions and questions, which I hope will help improve the manuscript.

Major comments:

As described in this manuscript, the established cELISA can efficiently differentiate C-strain/C-strain Erns immunized pigs. The PI value could represent the antibody level in vaccinated pig serum against C strain Erns protein, but could the PI value represent the efficiency of vaccination? Perhaps, this cELISA could only indicate the successful induction of antibody against Erns, it could not evaluate the neutralizing antibody titer induced by vaccination. Is it possible that high neutralizing antibody but low level antibody against Erns protein induced after vaccination.

Minor comments:

1: The figures presented in this manuscript is not clear.

2: As shown in Fig 1, mAb 1504 could not recognize the reduced Erns protein, thus I think this antibody could only recognize Erns protein structurally. Thus, a immunofluorescence assay based on cell line expressing the protein is needed if possible.

Reviewer 2 Report

Classical swine fever (CSF) is a highly contagious viral disease of domestic and wild pigs, which can be controlled effectively by vaccination with C-strain vaccine. Hawever this vaccination do not have DIVA capability. In this study, autors developed a novel competitive enzyme-linked immunosorbent assay (cELISA) based 1on a C-strain Erns specific monoclonal antibody (mAb 1504), aiming to serologically measuring immune responses to C-strain vaccine in pigs, and finally to make the C-strain become a DIVA compatible vaccine.

Line 79, 83-91: Constantly referring to previous articles distracts the reader, maybe it would also be worth describing the methodology in this article.

Line 144: a Detailed description of the drawing should be included in the text when referring to the drawing, while under the drawing there should only be a general description

Line 163,164: Figure 2: The font used in the description in figures A and B should be of one size, the legend of figure B is difficult to read from short to small

Reviewer 3 Report

The study by Wang et al. established a competitive ELISA to measure antibody response against CSFV C-strain vaccinated pigs. C-strain vaccine has been widely used in the field for CSF prevention and it’s a live attenuated virus made through serial passage in rabbits. Erns gene shares the most variability with field viral strains and current differentiation test rely on PCR or sequencing of Erns. This study generated a monoclonal antibody targeting vaccine strain Erns with specificity and further developed a cELISA and validated using a number of vaccinated/challenged samples.

Major comments:

-        C-strain vaccine information is lacking in the very beginning which made it difficult to understand the rational of study design. Please give brief introduction of the C-strain vaccine regarding its type, how it was made, efficacy, etc. though some descriptions appear in discussion part.

-        One of the reasons for this study to target on Erns maybe due to its low similarity with field CSFV strains. So, what’s the identity between C-strain Erns and field strain Erns in nt and AA levels?

-        And more critically, what’s the targeting epitope of the mAb 1504? Is that epitope immunogenic since the immunogenicity of the targeted region is the main consideration for cELISA development? This may be discussed by reviewing findings from other literatures.

-        In methods 2.3, please specify the number of serum samples for each source.
